# Distribution and Genomic Characterization of Third-Generation Cephalosporin-Resistant *Escherichia coli* Isolated from a Single Family and Home Environment: A 2-Year Longitudinal Study

**DOI:** 10.3390/antibiotics11091152

**Published:** 2022-08-25

**Authors:** Yin-Chih Feng, Ci-Hong Liou, Wailap Victor Ng, Feng-Jui Chen, Chih-Hsin Hung, Po-Yen Liu, Yu-Chieh Liao, Han-Chieh Wu, Ming-Fang Cheng

**Affiliations:** 1Department of Pediatrics, Kaohsiung Veterans General Hospital, Kaohsiung 813414, Taiwan; 2National Institute of Infectious Diseases and Vaccinology, National Health Research Institutes, Hsinchu 35053, Taiwan; 3Department of Biotechnology and Laboratory Science in Medicine, National Yang Ming Chiao Tung University, Taipei 112304, Taiwan; 4Department of Biological Science and Technology, National Yang Ming Chiao Tung University, Hsinchu 300093, Taiwan; 5Institute of Biotechnology and Chemical Engineering, I-Shou University, Kaohsiung 84001, Taiwan; 6Institute of Population Health Sciences, National Health Research Institutes, Miaoli 35053, Taiwan; 7School of Nursing, Fooyin University, Kaohsiung 83102, Taiwan; 8School of Medicine, National Yang Ming Chiao Tung University, Taipei 112304, Taiwan

**Keywords:** *Escherichia coli*, third-generation cephalosporin-resistant *Escherichia coli*, extended-spectrum β-lactamases, plasmid, AmpC β-lactamases, CMY-2, whole-genome sequencing

## Abstract

Third-generation cephalosporin-resistant *Escherichia coli* (CREC), particularly strains producing extended-spectrum β-lactamases (ESBLs), are a global concern. Our study aims to longitudinally assemble the genomic characteristics of CREC isolates from fecal samples from an index patient with recurrent CREC-related urinary tract infections and his family and swabs from his home environment 12 times between 2019 and 2021 to investigate the distribution of antibiotic resistance genes. CREC identified using the VITEK 2 were subjected to nanopore whole-genome sequencing (WGS). The WGS of 27 CREC isolates discovered in 137 specimens (1 urine, 123 feces, and 13 environmental) revealed the predominance of ST101 and ST131. Among these sequence types, *bla*_CTX-M_ (44.4%, *n* = 12) was the predominant ESBL gene family, with *bla*_CTX-M-14_ (*n* = 6) being the most common. The remaining 15 (55.6%) isolates harbored *bla*_CMY-2_ genes and were clonally diverse. All *E. coli* isolated from the index patient’s initial urine and fecal samples belonged to O25b:H4-B2-ST131 and carried *bla*_CTX-M-14_. The results of sequence analysis indicate plasmid-mediated household transmission of *bla*_CMY-2_ or *bla*_CTX-M-55_. A strong genomic similarity was discovered between fecal ESBL-producing *E. coli* and uropathogenic strains. Furthermore, *bla*_CMY-2_ genes were widely distributed among the CREC isolated from family members and their home environment.

## 1. Introduction

*Escherichia coli* is regarded as a top-ranked pathogen causing public health concerns and is widely distributed in both human and animal intestines. Most strains of *E. coli* are harmless, but some strains, which have acquired specific virulence attributes, are pathogenic and cause a variety of diseases in humans. Pathogenic *E. coli* strains are classified into categories based on the production of virulence factors and the site of infection. There are the following six well-described categories: enteroaggregative *E. coli* (EAEC), enterohemorrhagic *E. coli* (EHEC), enteroinvasive *E. coli* (EIEC), enteropathogenic *E. coli* (EPEC), enterotoxigenic *E. coli* (ETEC), extraintestinal *E. coli* (ExPEC), and Shiga toxin-producing *E. coli* (STEC) based on clinical characteristics and virulence factors [1]. The worldwide emergence of multidrug-resistant *E. coli* isolated from human, animal, and animal-derived products warrants global surveillance [2,3,4].

The worldwide prevalence of third-generation cephalosporin-resistant *Escherichia coli* (CREC), particularly that of strains carrying extended-spectrum β-lactamase (ESBL)-encoding *bla*_CTX-M_ or *bla*_AmpC_ genes, has dramatically increased [4,5]. A recent large-scale systemic review analyzing 29,872 participants from several countries discovered an 8-fold increase in the intestinal carriage rate of ESBL-producing *E. coli* in the community from 2000 to 2020 [6]. The fecal carriage rate of ESBL-producing *E. coli* varies globally; the highest carriage rate was observed in Southeast Asia (27%), whereas the lowest rate was reported in Europe (6%) over the last 20 years [6]. Some studies conducted in Taiwan have discovered trends paralleling the global increase in the prevalence of ESBL-carrying *E. coli* [7,8,9,10,11]. As in other East Asian regions, the dominant ESBL-producing *E. coli* strain in southern Taiwan is O25b-ST131, which produces CTX-M-14 [7,8,9,10]. Plasmid-mediated transmission has facilitated the rise in the prevalence of ESBL-carrying *E. coli* among healthy humans [12]. Broad-spectrum antimicrobials and antimicrobial abuse by humans and livestock exacerbate the spread of antibiotic resistance genes (ARGs) [13]. Similar to the prevalence of CTX-M-producing ST131 strains, the prevalence of CMY-2-producing *E. coli* seems to be increasing among humans [5]. Although the prevalence of ST131 producing CTX-M in human feces is well documented, few studies have investigated the fecal carriage rate of AmpC β-lactamase–producing strains, especially harboring *bla*_CMY-2_ [14,15].

The Oxford Nanopore MinION platform, which allows both DNA and RNA sequencing with a long read length, is ideal for whole-genome sequencing (WGS) and systematic characterization of bacterial isolates. Although the Oxford Nanopore MinION platform has low accuracy (85–92%) [16], a pipeline for polishing nanopore sequencing reads, generating consensus sequences, and correcting homopolymer errors was developed to attain an accuracy of >99.999% (Q50) [17]. Since WGS provides more comprehensive and cost-effective results, it is more effective than conventional polymerase chain reaction (PCR) amplification in terms of surveillance of ARGs and subtyping [18,19].

Our study aims to prospectively evaluate the longitudinal distribution and genomic characterization of CREC isolates from the household environment of an index patient with recurrent urinary tract infections (UTIs) caused by CREC.

## 2. Results

### 2.1. Participant Characteristic

The study enrolled a seven-member family, comprising the index patient with recurrent UTIs caused by CREC and his two parents, an aunt, an uncle, and two cousins. They lived together in their three-story house, had close contact with each other, shared the same gate entrance, and ate together in the same dining room. The index patient and his parents lived on the same floor, and his aunt, uncle, and two cousins lived on different floors. 

The questionnaire results are summarized in Table 1. All of the family members maintained good personal hygiene, had meals together, drank the same source of boiled tap water, and had similar exposure to a pet dog. Pork, fish, and chicken were the three most-consumed kinds of meats by the family. The index patient started mixed feeding with breastmilk and formula milk at birth and complementary feeding with formula milk and additional foods, such as baby rice cereal, pureed meat, and fruit, at the age of 6 months.

### 2.2. Antimicrobial Susceptibility Profiles

As shown in Appendix A, pulsed-field gel electrophoresis (PFGE) of *E. coli* isolates from index patient’s each stool and urine samples during the two episodes of hospitalization for UTI revealed a similar pattern, so we only chose one out of two *E. coli* isolates from urine samples in two episodes of UTI.

In addition to enrolling CREC isolates from one urine and two fecal samples from the index patient in separate hospitalizations, 121 *E. coli* isolates were collected from fecal samples and 13 *E. coli* isolates from environmental surface samples for CREC screening during the study period. In total, 27 of the 137 samples contained the following CREC isolates: 25 fecal samples, 1 urine sample, and 1 environmental surface sample. The three initial CREC isolates (I-01-S, I-02-S, and I-01-U, Table 2) from the index patients, collected during two separate hospitalizations, shared the same antimicrobial susceptibility patterns.

The CREC isolates demonstrated a range of susceptibility profiles against 17 classes of antimicrobials (Table 2). None of the isolates were susceptible to piperacillin, cefixime, or ceftriaxone. In total, 92.6%, 55.6%, 55.6%, 7.4%, 63.0%, 63.0%, 59.3%, and 55.6% of the isolates were non-susceptible to ampicillin/sulbactam, cefoxitin, ceftazidime, cefepime, ciprofloxacin, minocycline, gentamicin, and trimethoprim/sulfamethoxazole, respectively. In addition, 78.0% of the CREC isolates exhibited multidrug resistance (MDR). Nevertheless, all the isolates were susceptible to piperacillin/tazobactam, carbapenem, amikacin, and tigecycline.

### 2.3. Genome Sequences, Sequence Types, Serotypes, and Phylogenetic Groups

The genomes of the 27 CREC isolates belonged mostly to three phylogenetic groups (*n* = 9 in B1, *n* = 7 in D, and *n* = 6 in B2) and had a scattered distribution of sequence types (Figure 1A). As shown in Figure 1B, ST101 (25.9%, 7/27) was the most common sequence type, followed by ST131 (22.2%, 6/27). The pandemic sequence type ST131 was discovered in all B2 isolates (100%, 6/6), most of the C1-nM27 subclade (83.3%, 5/6), and most with the O25b:H4 serotype (66.7%, 4/6). Multilocus sequence typing (MLST) of the index patient’s isolates revealed a drift from the initial O25b:H4-B2-ST131 to O115:H31-B1-ST101 over the study period.

### 2.4. ARGs

#### 2.4.1. β-Lactamase-Encoding Genes and Household Transmission

WGS revealed that all the 27 CREC isolates contained an ESBL or other β-lactamase gene that confers cephalosporin resistance. As shown in Figure 2, the predominant ESBL gene type was *bla*_CTX-M_ (44.4%, *n* = 12), of which *bla*_CTX-M-14_ accounted for half (*n* = 6). Moreover, CTX-M was significantly more common among ST131 than the other sequence types (*p* < 0.001). Aside from ESBL genes, *bla*_TEM_ (74.1%, *bla*_TEM-1B_, *n* = 19, *bla*_TEM-135_, *n* = 1) was the most common gene type, followed by *bla*_AmpC_ (55.6% *bla*_CMY-2_, *n* = 15) among the other β-lactamase genes. The gene *bla*_CMY-2_ was significantly associated with the non-ST131 isolates (*p* = 0.03). Figure 2 displays the sequence types of CTX-M-producing and CMY-2-producing *E. coli*.

Plasmid-encoded *bla*_CMY-2_ and *bla*_CTX-M-55_ were detected in the genomes of the *E. coli* isolated from the family. Among these, *bla*_CMY-2_ was present in 15/27 (55.6%) of the CREC isolates from the family members, including 7/12 (58.3%) from the index patient, 3/4 (75%) from the aunt, 1/1 (100%) from one cousin, and 3/4 (75%) from another cousin. Moreover, *bla*_CMY-2_ was found in one environmental sample from the bathroom basin (Figure 3). Appendix A shows the genomic backgrounds of 15 *bla*_CMY-2_ plasmids. Using the National Center for Biotechnology Information Basic Local Alignment Search Tool (BLAST), the *bla*_CMY-2_-containing plasmids were analyzed and a high similarity was discovered in contigs (>99% identity) between some of them in two groups (CMY-2 plasmid 1 among *E. coli* isolates: E-01, I-08-S, I-11-S, I-09-S, I-10-S, I-07-S and CMY-2 plasmid 2 among *E. coli* isolates: I-06-S, I-05-S, and C2-02-S). Likewise, *bla*_CTX-M-55_ genes from different isolates (C2-03-S and A-04-S) shared similar plasmid sequences (Figure 4).

#### 2.4.2. ARGs Other Than β-Lactamase Genes

In addition to the aforementioned resistant β-lactamase genes, other ARGs were identified through WGS (Table 3). No carbapenem resistance gene was identified among the isolates. One isolate with a plasmid-mediated mobile colistin resistance gene, *mcr-1.1*, was discovered in one of the family members’ samples.

### 2.5. Virulence Factors (VF)-Encoding Genes

As shown in Table 4, all 27 CREC isolates harbored a wide variety of VF-encoding genes. Accordingly, 55.6% of the isolates were classified as extraintestinal pathogenic *E. coli* (ExPEC), 33.3% as uropathogenic *E. coli* (UPEC), and 40.7% as avian pathogenic *E. coli* (APEC). In addition, isolates with ST131 were more likely to be classified as ExPEC or UPEC than non-ST131 isolates (*p =* 0.02 and < 0.001, respectively).

## 3. Discussion

This study investigated fecal and urinary CREC isolates from a patient with recurrent UTIs and longitudinal collection of fecal samples from him and his asymptomatic family as well as samples from their home environment. To compare the genetics of these isolates, WGS was used for ARG and VF-encoding gene detection, phylogenetics, MLST, and serotyping.

Uniplex and multiplex PCR with specific primers were designed to investigate sequence types and characterize the antimicrobial resistance mechanisms of ESBL-carrying *E. coli* in the communities of southern Taiwan [7,8,9,10,20]. Traditionally, only certain sequence types, including ST69, ST73, ST95, and ST131, which are the major *E. coli* sequence types that cause community-acquired infection, were detected with target-specific primers [21]; O25b-ST131 was discovered to be the most common ESBL-producing *E. coli* clonal group in feces in our previous studies [8,20]. However, epidemiologic lineages might be underestimated due to a lack of corresponding primers. In this series, WGS was employed to overcome this limitation. Through nanopore sequencing, we found sequence types that might be neglected by traditional PCR, such as ST101 (25.9%, *n* = 7), which was even more than the pandemic ST131 (22.2%, *n* = 6). Similarly, WGS provided insight into the distribution of β-lactamase genes, including not only *bla*_CTX-M_ (44%), which we used to detect with specific primers in our previous work, but also *bla*_TEM_ (70.4%), *bla*_AmpC_ (55.6%), and numerous ARGs. 

WGS data revealed that fecal and urinary *E. coli* isolated from the index patient during the UTI had the following identical genetic features: both carried *bla*_CTX-M-14_ and ST131, and belonged to the B2 group, C1-nM27 subclade, and O25b:H4 serotype. This finding might suggest a potential relationship between intestinal colonization and UTIs. The digestive tract is a natural reservoir of *E. coli*, which may cause community infection and is a melting pot where resistance genes might be exchanged and cause resistant bacteria to rapidly increase under antimicrobial selective pressure. Among ESBL-producing *E. coli*, O25b:H4-B2-ST131 is a globally dominant clone characterized by simultaneous resistance to several classes of antimicrobials and contains numerous VF-encoding genes. In this study, all the ST131 clones were either ExPEC or UPEC. The combination of MDR and hypervirulence of O25b:H4-B2-ST131 *E. coli* represents a potential challenge to public health. 

As displayed in Figure 4A, we found the identical plasmids with *bla*_CMY-2_ genes were mostly from index patients, which persistently exist for over 7 months, and from one environmental surface during the same collection, which might be associated with contaminated waste. Moreover, plasmid-mediated horizontal gene transfer plays a crucial role in the dissemination of antimicrobial resistance [12], which is shown in Figure 4B,C. Plasmid contigs containing *bla*_CMY-2_ that were isolated from the index patient and one of his cousins had high sequence similarity (>99% identity), which might suggest horizontal gene transfer or acquisition from common sources. In addition, identical plasmid-mediated *bla*_CTX-M-55_ genes were discovered among isolates from the family, which might indicate the same potential phenomena.

According to Reuland et al. (2015), the prevalence of CMY-2-producing *E. coli* is lower than that of CTX-M-producing strains in human carriers [14]. CMY-2-producing *E. coli* are prominent in poultry, with a prevalence of 30% among CREC isolates [22]; such strains should not be overlooked because of the possible dissemination of ARGs to human-borne *E. coli* through plasmid-mediated transmission after food consumption. Only approximately 1% of Europeans harbor *E. coli* with *bla*_CMY-2_ [22,23,24]. However, recent studies, mainly in Asia, have indicated an increasing prevalence of CMY-2-producing *E. coli* [25]. Yan et al. (2004) discovered the community spread of *bla*_CMY-2_, with isolates harboring the gene found in the feces of food animals, retail ground meat, and the urine of patients with UTIs in Taiwan [26]. According to the results of the 7-year Study for Monitoring Antimicrobial Resistance Trends, the prevalence of CMY-2-producing *E. coli*, which causes intra-abdominal infection or UTI, had increased to 29.3% in Taiwan by 2014 [25].

In the present study, CMY-2-producing CREC was isolated from several family members, which might suggest long-term fecal colonization, especially for the index patients, and horizontal plasmid transfer between *E. coli* strains in southern Taiwan. However, the results of ARGs should be cautiously applied to general epidemiologic data due to the small sample size in this series. In addition, few studies have observed nonclonal dissemination of CMY-producing *E. coli* because the diversity of clonal lineages is high [27]. As shown in Figure 2, our CMY-2-producing *E. coli* isolates were genetically diverse, and ST101 was the most common (25.9%) among the nine sequence types found in the series. 

As we know, AmpC β-lactamase leads to resistance to third-generation cephalosporins, especially to cephamycin [28]. On the basis of antimicrobial stewardship, a previous study proposed cephamycin as a candidate to replace carbapenem for treating ESBL-carrying *E. coli* infection [29]. The rising prevalence of CMY-2-producing *E. coli* in feces presents several public health concerns. If harboring *bla*_CMY-2_, the pathogenic *E. coli*, such as ExPEC or UPEC, might eventually develop cephamycin resistance, which would limit the therapeutic options for MDR isolates, especially when an ESBL-producing strain is present [30]. Moreover, Drinkovic et al. (2015) reported that, although the prevalence of plasmid-mediated AmpC β-lactamase among *E. coli* was only 0.4% among urine isolates from an Auckland community, it still could affect the UTI treatment options [31]. 

Carbapenem remains the drug of choice for severe infections caused by ESBL-producing *E. coli* [32]; this is reasonable given that the susceptibility results of our series revealed no carbapenemase genes in any of the 27 CREC isolates. However, Chia et al. (2009) discovered that the concurrence of *bla*_CMY-2_ and porin deficiency led to carbapenem resistance, even in the absence of carbapenemase [33]. Thus, the emergence of isolates with *bla*_CMY-2_ genes may potentially suggest a greater distribution of MDR *E. coli*. Therefore, continued surveillance of both CTX-M-producing and AmpC-producing *E. coli* and other resistance mechanisms is warranted.

In our previous study, the prevalence of the fecal carriage of *mcr-1*-positive *E. coli* was low (2.4%) among community children in southern Taiwan [34]. Similarly, the prevalence in the present study was low, with *mcr-1*-positive *E. coli* encountered in only one of 27 CRECs from the 123 *E. coli* isolates. 

This study has some limitations. First, because of the small sample size and the few colonies chosen per fecal sample, this study may not represent the general genomic background of CREC isolates in the feces of individuals in Taiwan. Although we ensured that all the *E. coli* isolates were chosen based on colony morphology and were validated by MALDI-TOF MS to reduce selection bias, the number of colonies directly influences the ability to accurately characterize *E. coli* strain diversity [35]. Instead, new approaches such as qPCR or real-time PCR might improve the sensitivity of detection [36] and will be applied to describe the *E. coli* population structure in future studies. Next, although our results revealed the abundance and distribution of ARGs in a single family, we could not explain the transmission network of ARG-harboring *E. coli.* Then, the MLST of CREC isolates from the index patient showed a drift from ST131 to ST101 in the study period, but only a few CREC isolates from family members had the same ST101 clone. More bioinformation, from CREC and non-CREC isolates, may be needed for a more comprehensive explanation. Our results address the emergence of CREC isolates producing either CTM-X or CMY-2 β lactamases that might cause difficult-to-treat infection and pose a threat to human health. Finally, we did not investigate the resistance mechanism of intrinsic efflux pumps, which requires more information from transcriptional analysis, overexpression of efflux pump-encoding genes, generation of mutants, for instance, leading to porin loss, etc., which are beyond the scope of our study, and are warranted in future studies. 

## 4. Materials and Methods

### 4.1. Participants and Sample Collection

An uncircumcised male infant delivered vaginally at full term without any underlying disease experienced his first UTI episode at the age of 13 days in December 2019 and recurrent infection in April 2020. This infant was enrolled as our index patient during his second hospitalization for his recurrent UTI at Kaohsiung Veterans General Hospital (KVGH) in April 2020. After obtaining consent from all the family members in the same living space to participate in this study, each urine and fecal specimen of the index patient was collected during the two episodes of hospitalization. In addition, fecal samples were collected simultaneously from his family members, and household surfaces 12 times between May 2020 and September 2021 at intervals of 30–60 days. Swabs were also collected from the following household surfaces during the period: toilet rims; bathroom basins; the kitchen sink; living room, master bedroom, children’s room, bathroom, and kitchen floors; toys. Drinking water was sampled. The ethics committee of KVGH (approval No. VGHKS19-CT3-20) granted study approval in 2019.

### 4.2. Questionnaire Design

Questionnaires were completed after the first specimen collection. The questionnaire included items on travel history, animal contact, dietary habits, cleaning habits, weekly food consumption, weekly probiotic use, and medical history. Weekly food consumption and probiotic use were quantified using a Likert-type scale.

### 4.3. Microbiological Analysis and Antimicrobial Susceptibility Testing

Each fecal sample was streaked on a CHROMagar ECC plate (CHROMagar, Paris, France) and incubated at 37 °C for 24 h. Up to two *E. coli* isolates based on colony morphology from each specimen were subsequently validated through matrix-assisted laser desorption/ionization–time-of-flight (MALDI-TOF) mass spectrometry (MS) and subjected to antimicrobial susceptibility testing with the VITEK 2 automated system and AST-N320 cards (bioMérieux, Marcy-l’Etoile, France). Aside from tigecycline and colistin, the breakpoints of antimicrobials were based on the M100-S30 Clinical and Laboratory Standards Institute (2020) standard [37]. Classifications of 18 antimicrobials are listed in Appendix A. Minimum inhibitory concentrations of tigecycline were interpreted using the US Food and Drug administration (FDA) tigecycline susceptibility breakpoints for Enterobacteriaceae (https://www.fda.gov/drugs/development-resources/tigecycline-injection-products, accessed on 15 July 2022 [38]), while susceptibility tests for colistin were not shown due to unreliable results based on VITEK 2 methodology [39]. The validation of the antimicrobial resistance phenotype was based on the broth microdilution test for colistin. *E. coli* isolates that were non-susceptible to three or more antimicrobials from different classes were defined as exhibiting MDR [40]. PFGE was performed to identify *E. coli* isolates from index patient’s initial urine and fecal samples [41]. Isolates resistant to third-generation cephalosporin were used for WGS.

### 4.4. Nanopore Sequencing and De Novo Assembly

Total genomic DNA from 27 CREC strains was extracted with the QIAamp PowerFecal Pro Kit (Qiagen, Hilden, Germany); subsequently, DNA fragments longer than 3 kb were enriched with KAPA Hyper Beads (Roche, Wilmington, MA, USA) according to the manufacturer’s recommended procedures. WGS was performed on an ONT MinION sequencer (Oxford Nanopore Technologies, Oxford, UK) as previously described [42]. Briefly, each DNA sample was tagged with a unique barcode by using Rapid Barcoding Kit 96 (Oxford Nanopore Technologies). The barcoded DNAs were mixed and sequenced on a primed MinION SpotON Flow Cell (FLO-106MIN). Genome sequences were assembled de novo through a sampling strategy as previously described [42], and the assembled sequences were polished with Homopolish [17].

### 4.5. Pathogenic Types of E. coli

WGS was applied to detect VF-encoding genes due to high concordance compared with conventional PCR tools [43,44]. Established criteria were used to classify pathogenic *E. coli* based on VF-encoding genes. Isolates were categorized as extraintestinal pathogenic *E. coli* (ExPEC) if they carried two or more of the five virulence genes *papAH* and/or *papC*, *sfa/focDE*, *afa/draBC*, *kpsM*
*II*, and *iutA* [45]; as uropathogenic *E. coli* (UPEC) if they carried three or more of the four virulence genes *chuA**, fyuA, vat*, and *yfcV* [46]; avian pathogenic *E. coli* (APEC) if they carried four or more of the five genes *hlyF**, iutA, iroN, iss*, and *ompT* [47].

### 4.6. Bioinformatics and Statistical Analysis

The assembled and polished genome sequences were subjected to bioinformatics analysis including MLST, serotyping, phylogenetic grouping, examination of acquired ARGs, and detection and typing of plasmids were performed using the Center for Genomic Epidemiology (http://www.genomicepidemiology.org/services/, accessed on 15 July 2022 [48]) MLST 2.0.4 [18], SeroTypeFinder 2.0.1 [19], ClemonTyper 1.0.0 [49], ResFinder 4.1 [50], PlasmidFinder 2.1 [51], and pMLST 2.0 [51], respectively. In addition, WGS for known VF-encoding genes was performed using VirulenceFinder 2.0 [43,52]. In silico PCR was performed to distinguish subclades within pathogenic sequence types, and the Type (Strain) Genome Server [53] was referenced to construct phylogenetic trees. Moreover, the National Center for Biotechnology Information BLAST was used to compare plasmid sequences, and BLAST Ring Image Generator (V0.95) was employed to draw plasmid maps [54]. Statistical analyses were performed using chi-square or Fisher’s exact tests.

## 5. Conclusions

We observed a strong genomic association between fecal ESBL-producing *E. coli* and uropathogenic strains. WGS may be useful in the analysis of epidemiological types and trends, characterization of antimicrobial resistance mechanisms, and detection of ARGs and VF-encoding genes. Although a single-family study may not be representative of the general genomic backgrounds of CREC isolates in the feces of individuals for Taiwan, some unique ARGs, including *bla*_CTX-M_ and *bla*_AmpC_ genes, were found. To our knowledge, this is the first study in Taiwan to suggest *bla*_CMY-2_ gene distribution among asymptomatic individuals and to investigate the genomic characteristics of CREC isolates through nanopore sequencing. Furthermore, a longitudinal nationwide multi-center investigation of other resistance mechanisms, such as transcriptional regulation of intrinsic efflux pumps, for both CREC and non-CREC isolates in the human carriage, household spread, and environmental exposure is warranted. 

## Figures and Tables

**Figure 1 antibiotics-11-01152-f001:**
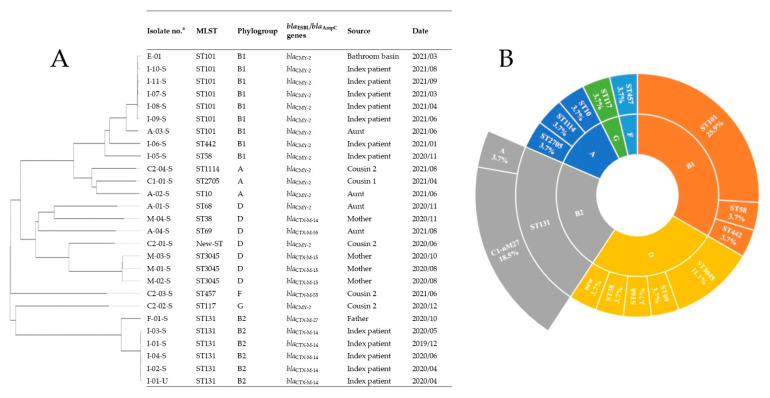
Multilocus sequence typing and phylogenetics traits of 27 third-generation cephalosporin-resistant *Escherichia coli* (CREC) isolates. (**A**) Phylogenetic tree of 27 CREC isolates produced through genome-wide phylogenetic analysis of the chromosomal sequences using the Type (Strain) Genome Server (https://tygs.dsmz.de/, accessed on 21 March 2022). (**B**) Proportions of sequence types (middle ring), phylogenetic groups (inner ring), and subclades (outer ring) of the pandemic ST131 strains predicted from the genome sequences of the 27 CREC isolates. ^a^ The S and U in isolation no. indicate the stool and urine origins, respectively.

**Figure 2 antibiotics-11-01152-f002:**
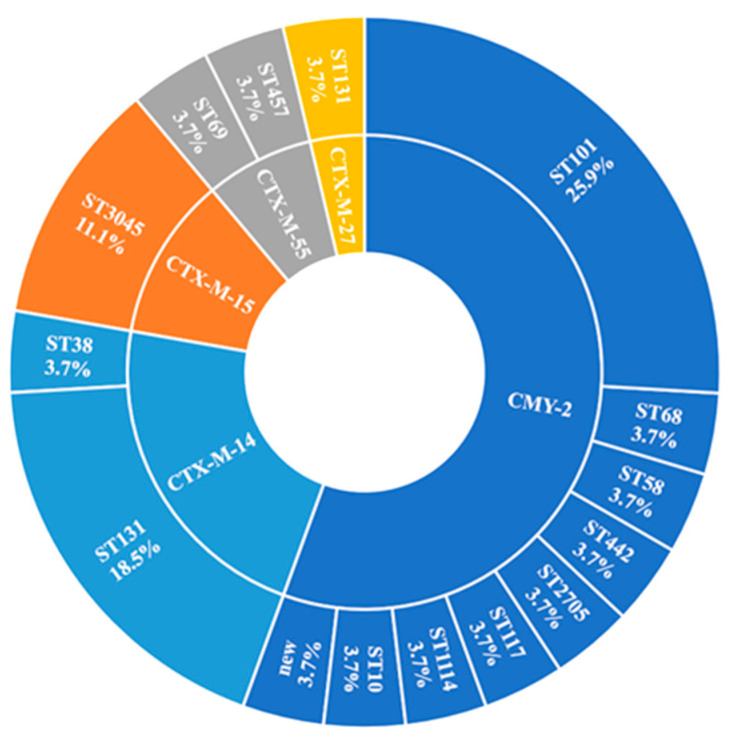
Relative abundance of sequence types among CTX-M-14-producing and CMY-2-producing *Escherichia coli*. The majority of CTX-M-14-producing and CMY-2-producing isolates had ST131 and ST101, respectively. The β-lactamases and sequence types of the 27 third-generation cephalosporin-resistant *E. coli* isolates were determined using MLST 2.0.4 for sequence type identification and ResFinder 4.1 for CTX-M/AmpC β-lactamase gene detection.

**Figure 3 antibiotics-11-01152-f003:**
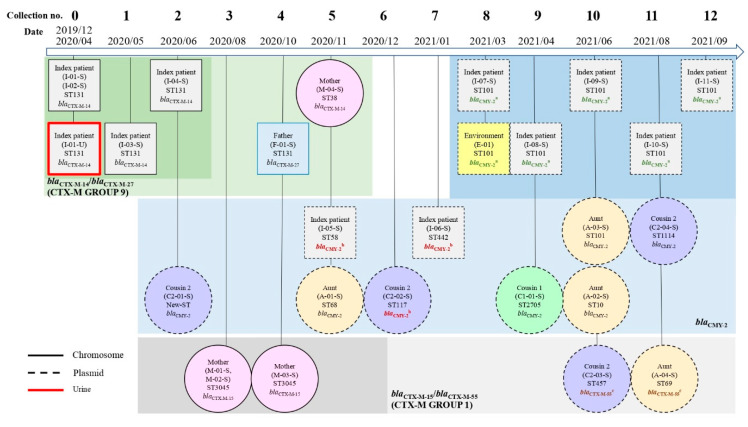
Sequence types and third-generation cephalosporin resistance genes among third-generation cephalosporin-resistant *Escherichia coli* (CREC) isolates. Distribution of sequence types and *bla*_ESBL_ and *bla*_AmpC_ genes in CREC isolated from family members and household environment from 12 collections between April 2020 and September 2021 after the index patient’s first urinary tract infection in December 2019. The isolate numbers are indicated in parentheses with the sequence types and antibiotic resistance genes. Characterization of β-lactamase-encoding genes and household transmission. Those β-lactamase-encoding genes with the same superscripts(^a,b,c^) are from similar plasmid sequences.

**Figure 4 antibiotics-11-01152-f004:**
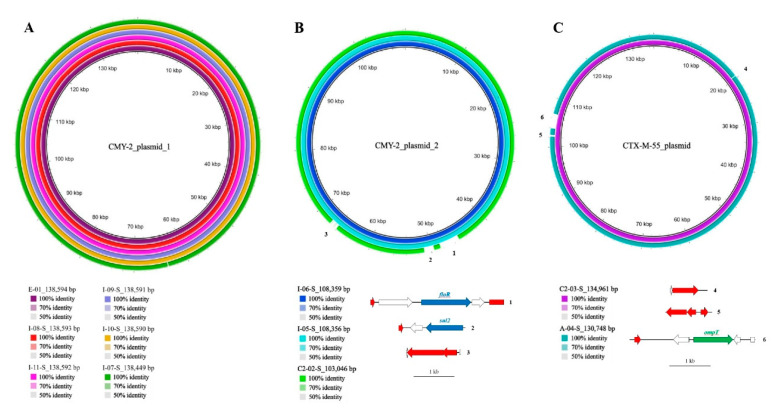
β-lactamase-encoding plasmids are highly similar among isolates. Concentric maps showing sequence similarities of (**A**) CMY-2 plasmid 1 among E-01, I-08-S, I-11-S, I-09-S, I-10-S, I-07-S; (**B**) CMY-2 plasmid 2 among I-06-S, I-05-S, C2-02-S; (**C**) CTX-M-55 plasmid among C2-03-S and A-04-S (innermost to outermost circles). Plasmid alignment was performed using the Basic Local Alignment Search Tool Ring Image Generator with the longest plasmid contigs as the reference sequence. The numbers 1–6 highlight the regions missing in some of the plasmids and comprise mobile elements (red arrow), antibiotic resistance genes (blue arrow), virulence factor–encoding genes (green arrow), and miscellaneous elements (white arrow). Colors and their shades indicate sequence similarities.

**Table 1 antibiotics-11-01152-t001:** Characteristics of index patient with recurrent urinary tract infection (UTI) and his six family members.

Variables	Index Patient	Mother	Father	Aunt	Uncle	Cousin 1	Cousin 2
**Demographic data**							
Age (years)	0.3	35	37	46	42	8	6
Male/Female	M	F	M	F	M	F	F
Travel abroad in the past 12 months	-	-	-	Japan	Japan China	Japan	Japan
Postal code	81146, Nanzi District, live in the same house but different floors						
Daily animal contact	Dog						
**Diet habit**							
Feed water source	RO water	RO water	RO water	RO water	RO water	RO water	RO water
Boiled water use	+	+	+	+	+	+	+
Vegetarian	Breastmilk or formula milk	-	-	-	-	-	-
Consumption of meats, consumption days per week							
Pork	0	6	7	5	6	6	5
Beef	0	1	1	1	1	1	1
Sheep	0	0	1	0	0	0	0
Chicken	0	4	2	4	4	4	4
Duck	0	1	2	1	1	1	1
Goose	0	0	1	0	0	0	0
Fish	0	4	7	4	4	4	4
**Family cleaning habits**							
Clean toilet bowl every month	+	+	+	+	+	+	+
Clean bathroom sink every month	+	+	+	+	+	+	+
Clean bathroom floor every month	+	+	+	+	+	+	+
Clean children’s bedrooms every month	+	+	+	+	+	+	+
Clean kitchen floor every month	+	+	+	+	+	+	+
Clean living room floor every month	+	+	+	+	+	+	+
Clean master bedroom every month	+	+	+	+	+	+	+
Clean toys every year	+	+	+	+	+	+	+
**Medical history**							
Previous hospitalization in the past 12 months	2019/12 UTI 2020/04 UTI	2019/12 Labor	-	-	-	2019/5 UTI	2019/07 Pneumonia
Previous antimicrobial treatment in the past 12 months	+	+	-	-	-	+	+
Previous probiotics use in the past 12 months, consumption days per week	0	0	0	1	0	7	7

M—male; F—female; RO—reverse osmosis; UTI—urinary tract infection.

**Table 2 antibiotics-11-01152-t002:** Antimicrobial resistance phenotypes of the 27 CREC isolates analyzed using VITEK 2 system and AST-N320 cards.

Antimicrobial Resistance Phenotype				Ampicillin/ Sulbactam	Piperacillin	Piperacillin/ Tazobactam	Cefazolin	Cefoxitin	Cefixime *^b^*	Ceftazidime *^b^*	Ceftriaxone *^b^*	Cefepime	Ertapenem	Imipenem	Amikacin	Gentamicin	Ciprofloxacin	Minocycline	Tigecycline	Colistin	Trimethoprim/ Sulfamethoxazole	Chloramphenicol	Azithromycin
Isolate No. *^a^*	Source	Date	*bla*_ESBL_/*bla*_AmpC_ Genes																				
I-01-S	Index case	2019/12	*bla* _CTX-M-14_																	ND *^c^*		ND *^d^*	ND *^d^*
I-01-U	Index case	2020/04	*bla* _CTX-M-14_																				
I-02-S	Index case	2020/04	*bla* _CTX-M-14_																				
I-03-S	Index case	2020/05	*bla* _CTX-M-14_																				
I-04-S	Index case	2020/06	*bla* _CTX-M-14_																				
I-05-S	Index case	2020/11	*bla* _CMY-2_																				
I-06-S	Index case	2021/01	*bla* _CMY-2_																				
I-07-S	Index case	2021/03	*bla* _CMY-2_																				
I-08-S	Index case	2021/04	*bla* _CMY-2_																				
I-09-S	Index case	2021/06	*bla* _CMY-2_																				
I-10-S	Index case	2021/08	*bla* _CMY-2_																				
I-11-S	Index case	2021/09	*bla* _CMY-2_																				
F-01-S	Father	2020/10	*bla* _CTX-M-27_																				
M-01-S	Mother	2020/08	*bla* _CTX-M-15_																				
M-02-S	Mother	2020/08	*bla* _CTX-M-15_																				
M-03-S	Mother	2020/10	*bla* _CTX-M-15_																				
M-04-S	Mother	2020/11	*bla* _CTX-M-14_																				
A-01-S	Aunt	2020/11	*bla* _CMY-2_																				
A-02-S	Aunt	2021/06	*bla* _CMY-2_																				
A-03-S	Aunt	2021/06	*bla* _CMY-2_																				
A-04-S	Aunt	2021/08	*bla* _CTX-M-55_																				
C1-01-S	Cousin 1	2021/04	*bla* _CMY-2_																				
C2-01-S	Cousin 2	2020/06	*bla* _CMY-2_																				
C2-02-S	Cousin 2	2020/12	*bla* _CMY-2_																				
C2-03-S	Cousin 2	2021/06	*bla* _CTX-M-55_																				
C2-04-S	Cousin 2	2021/08	*bla* _CMY-2_																				
E-01	Bathroom basin	2021/03	*bla* _CMY-2_																				
Percentage of non-susceptibility				92.6%	100.0%	0	100.0%	55.6%	100.0%	55.6%	100.0%	7.4%	0.0%	0.0%	0.0%	59.3%	63.0%	63.0%	0.0%	NA	55.6%	NA	NA

*^a^* S and U indicate fecal (stool) and urinary origin, respectively; *^b^* Third-generation cephalosporin; *^c^* Valid antibiotic resistance phenotype based on broth microdilution test for colistin, as suggested by M100-S30 Clinical and Laboratory Standards Institute (2020) standard; *^d^* No minimum inhibitory concentration available in VITEK 2 AST-N320 cards. Colored boxes: not susceptible; blank boxes: susceptible; ND-not detected; NA-not available.

**Table 3 antibiotics-11-01152-t003:** Antibiotic resistance genes (ARGs) in the CREC isolates predicted from their genome sequences.

Antimicrobial Resistance Genes			β-lactam							Aminoglycoside		Quinolone			Tetracycline			Colistin	Trimethoprim/Sulfamethoxazole							Amphenicol			Macrolide
Isolate No. *^a^*	Source	Date	*bla* _TEM_		*bla* _CTX-M_ * ^b^ *				*bla_AmpC_ ^b^*										*dfrA*				*sul*						
			*TEM-1B*	*TEM-135*	*CTX-M-14*	*CTX-M-15*	*CTX-M-27*	*CTX-M-55*	*CMY-2*	*aac(3)-IId*	*aac(3)-VIa*	*gyrA ^c^*	*qnrS1*	*qnrS13*	*tet*(A)	*tet*(B)	*tet*(M)	*mcr-1.1*	*dfrA5*	*dfrA12*	*dfrA14*	*dfrA17*	*sul1*	*sul2*	*sul3*	*cmlA1*	*catA1*	*floR*	*mph*(A)
I-01-S	Index case	2019/12																											
I-01-U	Index case	2020/04																											
I-02-S	Index case	2020/04																											
I-03-S	Index case	2020/05																											
I-04-S	Index case	2020/06																											
I-05-S	Index case	2020/11																											
I-06-S	Index case	2021/01																											
I-07-S	Index case	2021/03																											
I-08-S	Index case	2021/04																											
I-09-S	Index case	2021/06																											
I-10-S	Index case	2021/08																											
I-11-S	Index case	2021/09																											
F-01-S	Father	2020/10																											
M-01-S	Mother	2020/08																											
M-02-S	Mother	2020/08																											
M-03-S	Mother	2020/10																											
M-04-S	Mother	2020/11																											
A-01-S	Aunt	2020/11																											
A-02-S	Aunt	2021/06																											
A-03-S	Aunt	2021/06																											
A-04-S	Aunt	2021/08																											
C1-01-S	Cousin 1	2021/04																											
C2-01-S	Cousin 2	2020/06																											
C2-02-S	Cousin 2	2020/12																											
C2-03-S	Cousin 2	2021/06																											
C2-04-S	Cousin 2	2021/08																											
E-01	Bathroom basin	2021/03																											
Percentage of positive by either of one gene			20 (74.1%)		12 (44.4%)				15 (55.6%)	16 (59.2%)		19 (70.3%)			21 (77.8%)			1 (3.7%)	15 (55.6%)				17 (63%)			16 (59.2%)			8 (29.6%)

*^a^* S and U indicate fecal (stool) and urinary origin, respectively; *^b^* Genes related to third-generation cephalosporin resistance; *^c^* Point mutation of *gyrA* (S83L). Colored boxes: Presence of predicted ARG. Antimicrobial resistance phenotype and corresponding ARGs are indicated with the same color of box.

**Table 4 antibiotics-11-01152-t004:** Virulence factor–encoding genes among the 27 isolates of CREC.

Virulence Factor (VF)-Encoding Genes			Adherence												Capsule			Iron Uptake							Toxin													Miscellaneous												Definition		
Isolate No. *^a^*	Source	Date	*afaA*	*afaB*	*afaC*	*afaD*	*focC*	*hra*	*iha*	*lpfA*	*papA*	*papC*	*sfaD*	*yfcV*	*kpsE*	*kpsM II*	*kpsM III*	*chuA*	*fyuA*	*iroN*	*irp2*	*iucC*	*iutA*	*sitA*	*air*	*astA*	*ccI*	*cea*	*cib*	*cma*	*cnf1*	*hlyF*	*pic*	*sat*	*senB*	*tsh*	*vat*	*cvaC*	*eilA*	*etsC*	*gad*	*iss*	*katP*	*mcbA*	*mchF*	*ompT*	*terC*	*traT*	*usp*	ExPEC	UPEC	APEC
I-01-S	Index patient	2019/12																					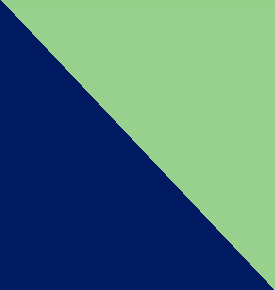																													
I-01-U	Index patient	2020/04																					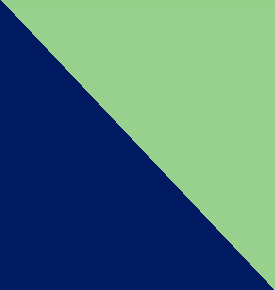																													
I-02-S	Index patient	2020/04																					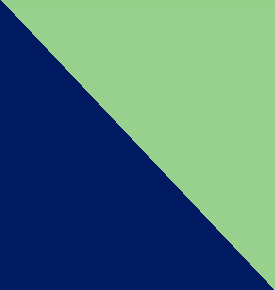																													
I-03-S	Index patient	2020/05																					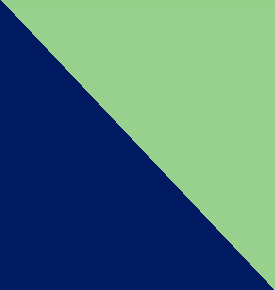																													
I-04-S	Index patient	2020/06																					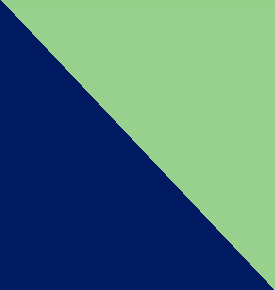																													
I-05-S	Index patient	2020/11																					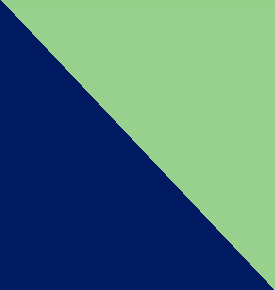																													
I-06-S	Index patient	2021/01																																																		
I-07-S	Index patient	2021/03																					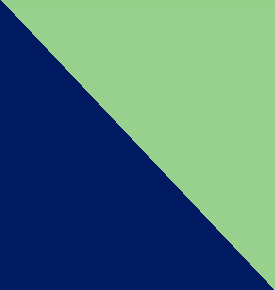																													
I-08-S	Index patient	2021/04																					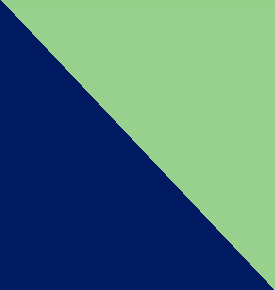																													
I-09-S	Index patient	2021/06																					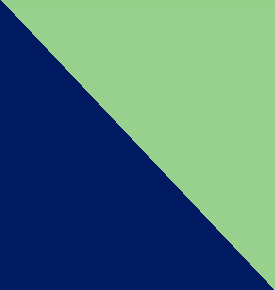																													
I-10-S	Index patient	2021/08																					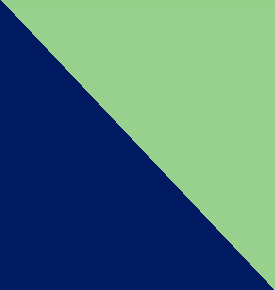																													
I-11-S	Index patient	2021/09																					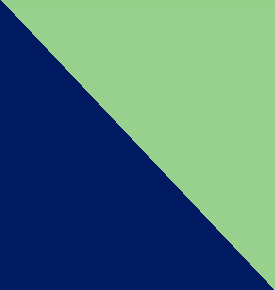																													
F-01-S	Father	2020/10																																																		
M-01-S	Mother	2020/08																																																		
M-02-S	Mother	2020/08																																																		
M-03-S	Mother	2020/10																																																		
M-04-S	Mother	2020/11																																																		
A-01-S	Aunt	2020/11																																																		
A-02-S	Aunt	2021/06																																																		
A-03-S	Aunt	2021/06																																																		
A-04-S	Aunt	2021/08																					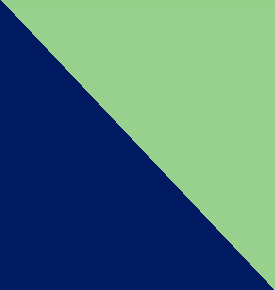																													
C1-01-S	Cousin 1	2021/04																																																		
C2-01-S	Cousin 2	2020/06																					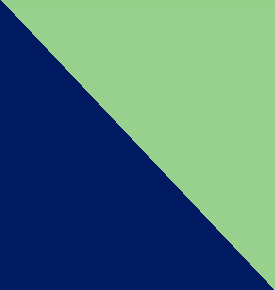																													
C2-02-S	Cousin 2	2020/12																					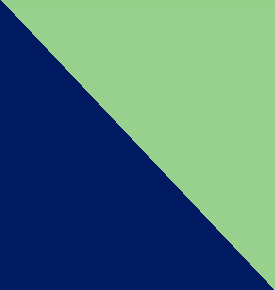																													
C2-03-S	Cousin 2	2021/06																					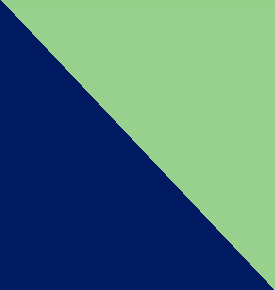																													
C2-04-S	Cousin 2	2021/08																																																		
E-01	Bathroom basin	2021/03																					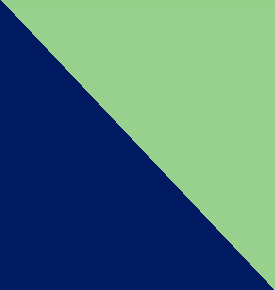																													
	ExPEC:≧2 of 5 markers, including *papAH* and/or *papC*, *sfa/focDE*, *afa/draBC*, *kpsM II*, and *iutA*																																																			
	UPEC:≧3 of 4 markers, including *chuA*, *fyuA*, *vat*, and *yfcV*																																																			
	APEC:≧4 of 5 markers, including *hlyF*, *iutA*, *iroN*, *iss*, and *ompT*																																																			
	Presence of predicted VF-encoding genes																																																			

*^a^* S and U indicate fecal (stool) and urinary origin, respectively. Isolates were categorized as extraintestinal pathogenic *E. coli* (ExPEC, blue boxes) if they contained ≥2/5 virulence markers (*papAH* and/or *papC*, *sfa/focDE*, *afa/draBC*, *kpsM II*, and *iutA*), as uropathogenic *E. coli* (UPEC, orange boxes) if they contained ≥3/4 virulence markers (*chuA*, *fyuA*, *vat*, and *yfcV*), and as avian pathogenic *E. coli* (APEC, green boxes) if they contained ≥4/5 markers (i.e., *hlyF*, *iutA*, *iroN*, *iss*, and *ompT*).

## Data Availability

The datasets generated during and/or analyzed during the current study are available from the corresponding author on reasonable request.

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
