# Peer review of "Distribution and Genomic Characterization of Third-Generation Cephalosporin-Resistant Escherichia coli Isolated from a Single Family and Home Environment: A 2-Year Longitudinal Study"

_antibiotics, 2022, doi:10.3390/antibiotics11091152_

Round 1

Reviewer 1 Report

This is a good report but I thought it could be more of a case report than research. Apart from that, it is a publishable paper after a few corrections indicated.   

Distribution and Genomic Characterization of Third Generation Cephalosporin Resistant Escherichia coli Isolated from a Single Family and Home Environment: A 2-Year Longitudinal Study

Abstract

Did not see the aim of the study in the abstract

Introduction/Results

Line 84 -85 why did you collect samples from the family who are not on the same floor as the index case or what were your criteria for enrolling the family members

Line 130- 132 it is not clear how ST131 drift to ST101 over the study period

Line 145-147 the sentence is not clear not sure which was predominantly CTXM or TEM

LINE 164-165; Can you specify which sample isolates have the high similarity?

Line 240 “The digestive tract is a natural reservoir of ESBL-producing E. coli,”  I think E.coli, not ESBL E.coli.

Table 4

Can you make table 4 clearer?

Was finding it difficult to locate the below markers

(ExPEC, dark gray boxes) virulence markers (papAH and/or papC, sfa/focDE, afa/draBC, kpsM II, and 212 iutA), 

uropathogenic E. coli (UPEC, medium gray)  ≥3/4 virulence markers (chuA, fyuA, vat, and yfcV),

avian pathogenic E. coli (APEC, light gray) ≥4/5 markers (i.e., hlyF, iutA, iroN, iss, and ompT)

Author Response

Dear Reviewer:

The authors appreciate your excellent review and have tried in our best ability to fulfill the revision points. We have made the appropriate changes in a point-by-point basis. Please see the attachment.  

Thank you again for your precious review.

Reviewer 2 Report

The figures showing the clonal relation ship with CTX and CMY genotypes related to sequence type is well done.  However, the sample collection does not permit establishment of the direction of transmission. The discussion of prevalence within a small family group, where 11 of 27 of the isolates were recovered from the same index infant, is not relevant. An important and valuable part of this manuscript is the characterization of the virulence and ARG profiles of the ST131 O25 b:H4 and the ST101 O115:H31 strains.

The index 2019-2020 UTI in an infant (ST131) including fecal carriage, and subsequent fecal carriage of ST101 does not seem to correlate with family. This begs the question of sources not included in the data collected. Why was there no isolate from the second UTI on 3/2021. We do not know that this E. coli isolate was even responsible for the UTI. Who prepared the formula and meals?  Were there other caregivers for the infant?   Was infection contracted in a nursery or from a midwife? How often were diapers changed?  The mother never yielded either ST101 or ST131. Or did they miss detecting it in the family by only collecting 2 samples from asymptomatic individuals. The sampling strategy needs to be better explained.

My biggest concern is selecting 2 isolates per fecal samples as representative of the total E. coli flora of that individual and small numbers of UPEC present in the GIT could be missed that could selectively colonize the urinary tract. We generally select a minimum of 6 isolates and that is to co confirm an infection while trying to represent normal flora would require many more.  This is a case were performing qPCR on the fecal samples to search for virulence factors and ARG might be a better approach to characterize the family microbiota rather than so few individual isolates.

Author Response

(The authors gave the same response as above.)

Reviewer 3 Report

This manuscript is noteworthy with a great interest for the scientific community. I support its possible publication after appropriate minor modifications as outlined below.

Line 26: “we longitudinally collected” - I strongly suggest to the authors to avoid the using of verbs in a personal mode form, it is not so scientifically sound. Please revise this concern throughout the manuscript.

Line 53: “Southeast Asia has exhibited” “Europe has exhibited” – not scientifically sound, the continents/regions cannot exhibit

Line 59: “antibiotics” – please change with antimicrobials, is a more appropriate term

Line 63: if the authors stated “few studies”, please add more reference at the end of the sentence (line 64)

Lines 70-73: unclear sentence, please rephrase or divide it

Line 74: (from age 4-21 months) – unclear

Lines 74-79: - please rephrase these lines, clearly defining the study aim. In this form seems to be materials and methods section

Line 79: within the introduction section, for a better understanding of the importance of E. coli for the public health, a clear definition of pathogenic E. coli is needed, mentioning the classification of pathotypes, especially of those able to produce toxins. Also, I suggest to authors to highlight the importance of the monitoring of E. coli within the food-chain, also via other animal origin foodstuffs (e.g. cheese - doi: 10.3390/antibiotics11060721 or beef - doi: 10.3390/foods9111543). This articles can be consulted and cited in order to increase the MDPI journals.

Line 96: please indicate and explain under the Table 1 the used abbreviations (e.g. M, F or RO)

Line 290: “E. coli. Therefore” instead of “E. coli; therefore

Line 294: please insert reference at the end of the sentence

Line 313: it would be important to obtain from each individual writing consent to participate in the study

Line 334: please indicate the total number of tested antimicrobials and the classes they belong to (you can include them in a separate supplementary file)

Lines 334-335: please ensure that in the mentioned CLSI guideline are available breakpoint for all of the antimicrobials that are included in the AST- N320 bioMérieux card

Line 336: “three or more antimicrobials from different classes” instead of “three or more categories of antimicrobials” seems to be a more appropriate definition for multidrug resistance

Line 357: in order to validate their results, the authors must refer to used positive and negative controls with their molecular biology investigations (especially PCR reactions)

Line 372: please replace “discovered” with “observed”

Line 377: within the conclusion section please highlight the study limitation(s) and approach future research directions

Line 406: in the reference list please bold the publication year of articles, and italicize the journal volume

Line 421: “Escherichia coli” – please italicize the scientific name of all species mentioned in the reference list

Author Response

(The authors gave the same response as above.)

Reviewer 4 Report

This manuscript by Feng et al. reports the distribution of ESBL and AmpC genes in E. coli isolates from a single family and home environment. The authors revealed the index patient had been carried third-generation cephalosporin resistant E. coli belonging to two different Sequence Type (ST131 and ST101) in a long term. In addition, the family members asymptomatically harbored ESBL/AmpC-producing E. coli with different genetic backgrounds. The experiment was rigidly done and the manuscript is well written based on the appropriate data. This reviewer asks the authors to consider the following minor points.

1.     The information for ESBL and AmpC genes should be added to Table 2 and Figure 1A for better understanding.

2.     Line 273. “clonally diverse” is a little confusing. “genetically diverse?”

3.   To lead the conclusion “ blaCMY-2 gene person to person transmission” clearly, the following points should be presented. 

a)     Ring images of the remaining 6 blaCMY-2 plasmids (A-03-S, C2-04-S, C1-01-S, A-02-S, A-01-S, C2-01-S)

b)     Summary table showing the genetic backgrounds (pMLST, Inc-type, and size) of 15 blaCMY-2 plasmids

Author Response

(The authors gave the same response as above.)

Round 2

Reviewer 2 Report

Thank you for including the limitations and editing the introduction and results so that it do not seem to promise something it does not deliver.